



# Greenland and Canadian Arctic ice temperature profiles

Anja Løkkegaard[1,2], Kenneth Mankoff[1], Christian Zdanowicz[3], Gary D. Clow[4], Martin P. Lüthi[5], Samuel Doyle[6], Henrik Thomsen[1], David Fisher[7], Joel Harper[8], Andy Aschwanden[9], Bo M. Vinther[10], Dorthe Dahl-Jensen[10], Harry Zekollari[11], Toby Meierbachtol[8], Ian McDowell[12], Neil Humphrey[13], Anne Solgaard[1], Nanna B. Karlsson[1], Shfaqat Abbas Khan[2], Benjamin Hills[14], Robert Law[15], Bryn Hubbard[6], Poul Christoffersen[15], Mylène Jacquemart[16], Robert S. Fausto[1], and William T. Colgan[1]

[1]Geological Survey of Denmark and Greenland, Copenhagen, Denmark
[2]DTU Space, Technical University of Denmark, Denmark
[3]Department of Earth Sciences, Uppsala University, Uppsala, Sweden
[4]Institute of Arctic and Alpine Research, University of Colorado Boulder, USA
[5]Geographical Institute, University of Zurich, 8052 Zurich, Switzerland
[6]Centre for Glaciology, Department of Geography and Earth Sciences, Aberystwyth University, Aberystwyth, SY23 3DB, UK
[7]Department of Earth and Environmental Sciences, University of Ottawa, Ottawa, Canada, K1N 6N5
[8]Department of Geosciences, University of Montana, Missoula, MT,59812, USA
[9]Arctic Region Supercomputing Center, University of Alaska Fairbanks, Fairbanks, AK, USA
[10]Centre for Ice and Climate, Niels Bohr Institute, University of Copenhagen, Copenhagen, Denmark, 2100
[11]Laboratory of Hydraulics, Hydrology and Glaciology (VAW), ETH Zurich, Zurich, Switzerand; Swiss Federal Institute for Forest, Snow and Landscape Research (WSL), Birmensdorf, Switzerland; Laboratoire de Glaciologie, Université libre de Bruxelles, Brussels, Belgium
[12]Graduate Program of Hydrologic Sciences, University of Nevada, Reno, Reno, NV, USA
[13]Department of Geology and Geophysics, University of Wyoming, Laramie, WY, USA
[14]Department of Earth and Space Sciences, University of Washington, Seattle, WA, USA
[15]Scott Polar Research Institute, University of Cambridge, Cambridge, UK
[16]Laboratory of Hydraulics, Hydrology and Glaciology (VAW), Department of Civil, Environmental and Geomatic Engineering, ETH Zurich, Zurich, Switzerand and Swiss Federal Institute for Forest, Snow and Landscape Research (WSL), Birmensdorf, Switzerland

**Correspondence:** Anja Løkkegaard (aloe@geus.dk)

**Abstract.** Here, we present a compilation of 85 ice temperature profiles from 79 boreholes from the Greenland Ice Sheet and peripheral ice caps, as well as local ice caps in the Canadian Arctic. Only 25 profiles (32%) were previously available in open-access data repositories. The remaining 54 profiles (68%) are being made digitally available here for the first time. These newly available profiles, which are associated with pre-2010 boreholes, have been submitted by community members or digitized from published graphics and/or data tables. All 85 profiles are now made available in both absolute (meters) and

normalized (0 to 1 ice thickness) depth scales, and are accompanied by extensive metadata. This metadata includes a transparent description of data provenance. The ice temperature profiles span 70 years, with the earliest profile being from 1950 at Camp VI, West Greenland. To highlight the value of this database in evaluating ice flow simulations, we compare the ice temperature profiles from the Greenland Ice Sheet with an ice flow simulation by the Parallel Ice Sheet Model (PISM). We find a cold bias

in modeled near-surface ice temperatures within the ablation area, a warm bias in modeled basal ice temperatures at inland



cold-bedded sites, and an apparent underestimation of deformational heating in high-strain settings. These biases provide process-level insight on simulated ice temperatures.

# 1 Introduction

With the tremendous social implications of sea-level change, the past decade has seen a proliferation of simulations of the
current and future geometry and dynamics of the Greenland ice sheet (Bindschadler et al., 2013; Goelzer et al., 2020; Aschwanden et al., 2021). These present-day complex ice flow models build upon the legacy of simpler past thermodynamic models (Letréguilly et al., 1991; Huybrechts et al., 1991; Funk et al., 1994; Calov and Hutter, 1996). Due to the high sensitivity of ice viscosity to ice temperature, the thermal state of the sheet is a critical element of these simulations (Colgan et al., 2015). At present, however, the englacial temperature fields of even cutting-edge ice-sheet simulations remain largely unevaluated
against observed temperatures (Aschwanden et al., 2019). While recent studies show potential for deriving internal ice temperatures from satellite or airborne data (Macelloni et al., 2019; Jezek et al., 2022), these techniques are not yet widely employed. There are consequently diverse opinions on Greenland's basal thermal state across the current generation of thermo-mechanical ice flow models (MacGregor et al., 2016).

Several different methods for measuring ice temperatures have been used on the Greenland Ice Sheet and in the Canadian
Arctic. The methods include: borehole logging where a temperature sensor is moved up or down the borehole measuring either "continuously" as the probe moves down or is stopped to measure at every depth known as "stop-and-go" (Johnsen et al., 1995; Clow, 2008); sensor strings where thermistors are frozen into the ice and ice temperatures are recorded at various depths at the same time (Iken et al., 1993; Ryser et al., 2014); and, fiber-optic distributed temperature sensing (DTS), where the ice temperature is measured near-continuously along the full cable length (Law et al., 2021). We summarize the methods and their
advantages and disadvantages in Table 1.

Ice temperature profiles collected in Greenland and the Canadian Arctic have not been systematically compiled into a coherent database. In particular, many pre-1990 ice temperature profiles languished in undigitized reports or gray literature. This presented a clear motivation to assemble ice temperature measurements into a consistent and comprehensive community resource. Here, we describe our compilation of ice temperature profiles from Greenland and the Canadian Arctic into an
open-access database with well-documented and uniform metadata for each entry. We include Canadian Arctic ice caps in our predominantly Greenland database, as these regions reside within the domain of some Greenland ice flow models (Tarasov and Peltier, 2004; Gowan et al., 2021).

The earliest temperature profile in our database is from 1950, when a 125 m profile was measured at Camp VI in West Greenland (Heuberger, 1954). Earlier temperature profiles may exist, and incorporating these profiles into the database is an
ongoing process. We restrict our database to ice temperature profiles extending well below the depth of the seasonal temperature cycle. At cold, dry sites, this is often approximated to be 10 to 15 m below the surface (Cuffey and Paterson, 2010). Similar to a recent effort to compile surface mass balance observations into a readily accessible common framework (Machguth et al., 2016), we aim to create a community resource that facilitates comparisons between simulated and observed ice temperatures.





Here, we describe the format of the database as well as the sources of our ice temperature data and metadata. We also discuss
best practices for comparing observed ice temperatures with simulated ice temperatures from a 3-D thermo-mechanical model.
Finally, we make an appeal to community colleagues to continually update this dataset, providing instructions for them to do
so.

**Table 1.** Advantages and disadvantages of the main methods for measuring deep ice temperatures. This compilation ignores the absolute
accuracy of the ice temperature sensor, which can vary greatly within any one method type.

| Method | Advantages | Disadvantages |
|---|---|---|
| **Borehole logger** | • recoverable and reusable, leaves the borehole empty<br>• continuous logging is possible over entire borehole depth | • some require 'stop-and-go' measurements to equilibrate at each measurement depth<br>• logger depth uncertainty between individual measurement depths<br>• measurements require on-site presence, and thus unattended time series, are not possible |
| **Thermistor string** | • relatively cheap instruments and data logger at the ice surface<br>• unattended ice profile measurements through time are possible<br>• voltage-based measurements are a relatively simple and established technology | • single deployment, cable generally not recovered when frozen in borehole<br>• cable resistance correction must be calculated for large ice depths<br>• number of measurements with depth limited as each thermistor requires a separate cable core |
| **Digital sensor string** | • relatively cheap instruments and data logger at the ice surface<br>• unattended ice profile measurements through time are possible<br>• large number of sensors with depth possible with common cable and data transfer protocols<br>• measurement is not affected by cable resistance or capacitance effects | • single deployment, cable generally not recovered when frozen in borehole<br>• detection of temperature change limited by resolution<br>• heat dissipation from electronics can affect measurements |
| **Fiber optic distributed sensing string** | • near-continuous vertical ice temperature profile<br>• unattended ice profile measurements through time are possible<br>• no inter-sensor uncertainty at different depths | • single deployment, cable generally not recovered when frozen in borehole<br>• sensitive and expensive data logger at ice surface<br>• relatively expensive compared to thermistor and digital sensor strings |





## 2 Temperature Data

The temperature database currently comprises a total of 85 temperature profiles. In two instances the same borehole was logged
twice at different times (Thomsen et al., 1991). Other boreholes which have been logged more than once also include GISP2-
D, GISP2-G, NGRIP-2, NGRIP-S2, NEEM-D, however, these are currently not included in the database. The database also
contains five partial profiles at one site (Lüthi et al., 2002). The boreholes were all situated less than 25 m from one another,
and are therefore regarded as one borehole, meaning that the 85 profiles are considered to originate from 79 unique boreholes.

See Figure 1 for an overview of the drill site locations, and the database for the complete list of borehole IDs. Borehole
IDs denote the general location of the borehole, project-specific borehole IDs if multiple boreholes exist at that location,
and the year of the measurement. Borehole temperature records were entered into the database in two different formats: A)
tabulated original measurements, provided either by community members or obtained from historical reports, or B) published
graphics of measured temperature profiles digitized using the publicly available digitization tool WebPlotDigitizer (https:
//automeris.io/WebPlotDigitizer; Last Access 3 March 2022; Rohatgi (2021)).

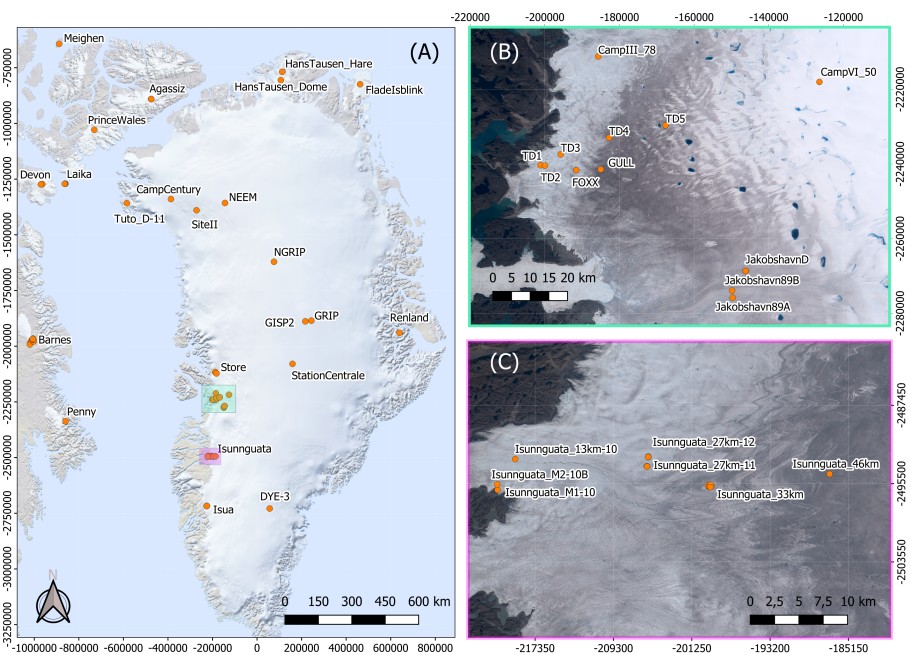

**Figure 1. (A)**: Overview of the drill site locations of temperature profiles contained in the database (plot created using QGreenland, Moon
et al. (2021)). Note, locations where several drill sites exist close to each other, only the common name is shown, e.g. Agassiz. The green
and purple box indicates regions where numerous profiles have been drilled. **(B)**: Zoomed view of the region near Jakobshavn glacier. **(C)**:
Zoomed view of the region near Isunnguata Sermia glacier. Panels **(B)** and **(C)** show positions plotted on top of Sentinel-2 multispectral
satellite imagery, 10 m resolution from 2019 (MacGregor et al., 2020). All maps are displayed in EPSG: 3413 polar stereographic projection,
with the Easting and Northing given in meters.





Overall, 61 of the 85 profiles were available as tabulated temperature measurements (Heuberger, 1954; Classen, 1977; Stauffer and Oeschger, 1979; Clarke et al., 1987; Thomsen et al., 1991; Iken et al., 1993; Hansson, 1994; Cuffey et al., 1995; Thomsen et al., 1996; Cuffey and Clow, 1997; Dahl-Jensen et al., 1998; Fischer et al., 1998; Lüthi et al., 2002; Kinnard et al., 2006; Buchardt and Dahl-Jensen, 2007; Kinnard et al., 2008; Lemark and Dahl-Jensen, 2010; Rasmussen et al., 2013; Ryser et al., 2014; Harrington et al., 2015; Hills et al., 2017; Zekollari et al., 2017; Doyle et al., 2018b; Hubbard et al., 2021a; Harper and Meierbachtol, 2021; Law et al., 2021). The remaining 24 profiles were digitized from figures (Hansen and Landauer, 1958; Davis, 1967; Paterson, 1968; Classen, 1977; Paterson et al., 1977; Colbeck and Gow, 1979; Gundestrup and Hansen, 1984; Blatter and Kappenberger, 1988; Gundestrup et al., 1993).

Here, we describe both the initial Dataverse database release (Mankoff et al., 2022) and the GitHub living repository (Mankoff, 2022). The Dataverse database (Mankoff et al., 2022) contains four curated files: two comma-separated value (CSV) files with temperature and depth-normalized temperature at each site, a KML file of borehole locations, and a metadata file. The GitHub living repository (Mankoff, 2022) is where raw data is collected, curated, and documented in order to create the final data. The repository is termed living, because it is also the entry point for new data into future database versions.

GitHub living repository (Mankoff, 2022) consists of high-level post-processed CSV files ready for use, and low-level folders with original sources and notes for each temperature profile. The CSV files include two data files combining all profiles, one file with depth in units of meters with a step size of 1 m, and another file with normalized depth as 0-1 with a step size of 0.01. This standardization of the profiles from coarse discrete measurements to uniformly finely spaced values was done by interpolating between points, using cubic spline interpolation with no overshoot. A third CSV file combines all metadata for each profile and, finally, a GIS file (KML format) provides borehole locations. Folders also contain a file named "data.csv" that is the original data at the original depth resolution, a README.org file with any relevant notes, including "cleaned" email correspondence when personal communications were involved in recovering the data, or other files detailing data acquisition.

Where ice temperature profiles are derived from a graphic (Barnes_(B4,D4)_1974, Barnes_T0(975_1976, 91_1977, 81_1978, 61_1977, 20_1979), Camp Century, Devon72, Devon73, DYE-3, Isua_10-14, Laika_75a-75e, Meighen67, SiteII, Tuto_D-11), the graphic is included in the folder. These folders also include a .tar or JSON file, which can be opened in WebPlotDigitizer, that contains the digitized profile and shows the digitization method. Graphical temperature data was digitized in the following way: The figure was loaded into WebPlotDigitizer as an xy-plot. The axis of the figure was then calibrated by defining two points on each axis, with these definition points preferably being as far away from each other as possible. The program has two options for digitizing, "manual" or "automatic" extraction, and typically a combination of both methods was used. Using the automatic mode, profiles were defined by creating a mask, and an automatic extraction was used to produce an initial set of points. The manual mode was then used to adjust these points where necessary to fine tune visual match to the graphic. For figures that contained both in-situ temperature points and a fitted temperature line, only the points were detected using WebPlotDigitizer's "blob detector" automatic algorithm.

Digitizing data from graphics can introduce non-trivial digitization uncertainty associated with potentially improper alignment of the defining axes (systematic uncertainty) and/or improper alignment of the data points (random uncertainty). The magnitude of this digitization uncertainty is proportional to the sizes of the graphic and data range, which vary from profile





to profile. To highlight the potential importance of digitization uncertainty, we propagate the uncertainty associated with a ±2 pixel error in each of the axes defining points for both temperature and depth in the digitized profiles with the thinnest ice (Tuto_D-11; 48 m) and the thickest ice (DYE-3; 2038 m). The Tuto_D-11 graphic (Davis, 1967)(Figure 2A) is 1,000 pixels tall and spans 75 m, which yields a ±2 pixel depth uncertainty of ±0.15 m. The Tuto_D-11 graphic is 600 pixels wide and spans 32 °C, which yields a ±2 pixel temperature uncertainty of ±0.1 °C. In comparison, the DYE-3 graphic (Gundestrup and

Hansen, 1984)(Figure 2B) is 1200 pixels tall, spans 2200 m and yields a ±2 pixel depth uncertainty of ±3.6 m. The DYE-3 graphic, 900 pixels wide, spans 8 °C and yields a ±2 pixel temperature uncertainty of ±0.02 °C. These end-member scenarios highlight how digitization uncertainty varies from graphic to graphic.

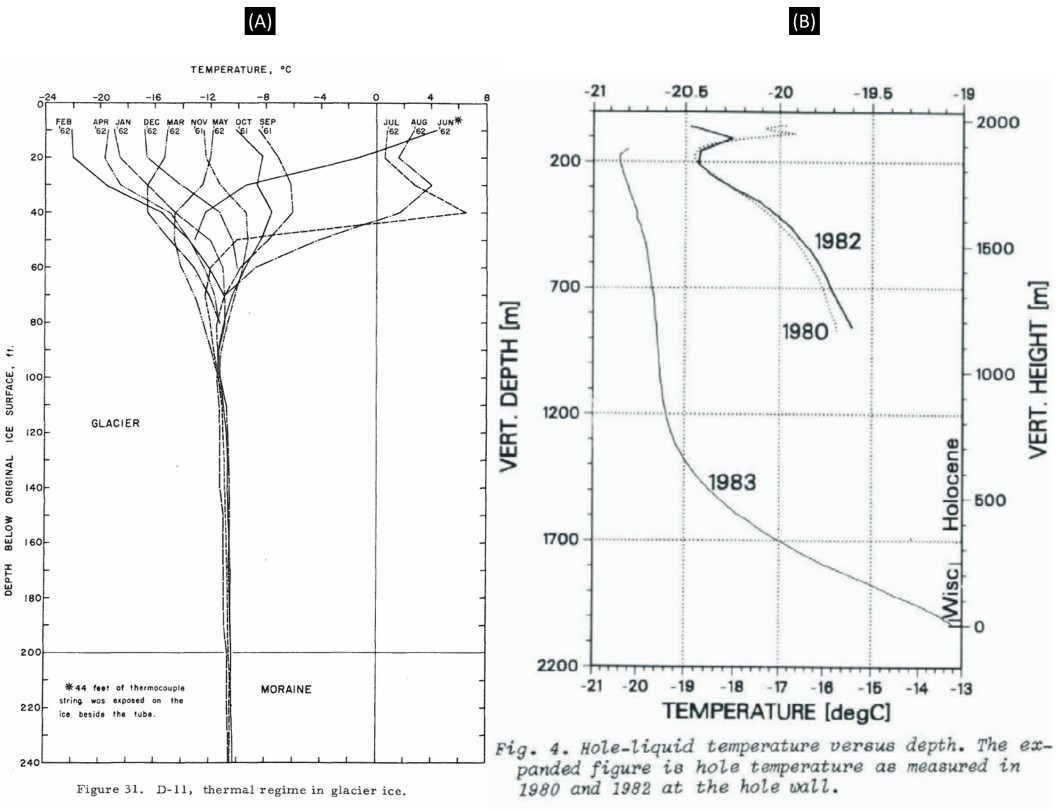

**Figure 2. (A)**: Graphic from which the Tuto_D-11 profile is digitized (Davis, 1967). This figure is reproduced from a US Government publication. **(B)**: Graphic from which the DYE-3 profile is digitized (Gundestrup and Hansen, 1984). This figure is reproduced with permission from the International Glaciology Society.

All ice temperature profiles – from both digitized and tabulated sources – were standardized by interpolating between points, using cubic spline interpolation with no overshoot, in order to resolve the two common depth scales: the absolute

depth scale at 1 m vertical resolution (Figure 3A, C, E) and the normalized depth scale at non-dimensional 0.01 vertical resolution (Figure 3B, D, F). The latter allows easy comparison between sites of different ice thicknesses, and is useful in





overcoming slight differences in ice thickness when comparing observed and modeled temperature profiles. The normalized depth scale temperature file is named "temperature_dnorm.csv". The absolute depth scale temperature file, which is expressed as temperature with depth from the surface, is named "temperature.csv". In each file, 'NaN' means that no temperature data is available at the given depth, and in the absolute depth file '-999' refers to an elevation below bedrock. Ice thicknesses vary significantly across the database profiles, from 3085 m at NGRIP to 48 m at Tuto_D-11, so the number of -999 below-bed null-values varies between sites. This is because the temperature vector length is constant across the database entries, but the ice thickness varies across database entries.

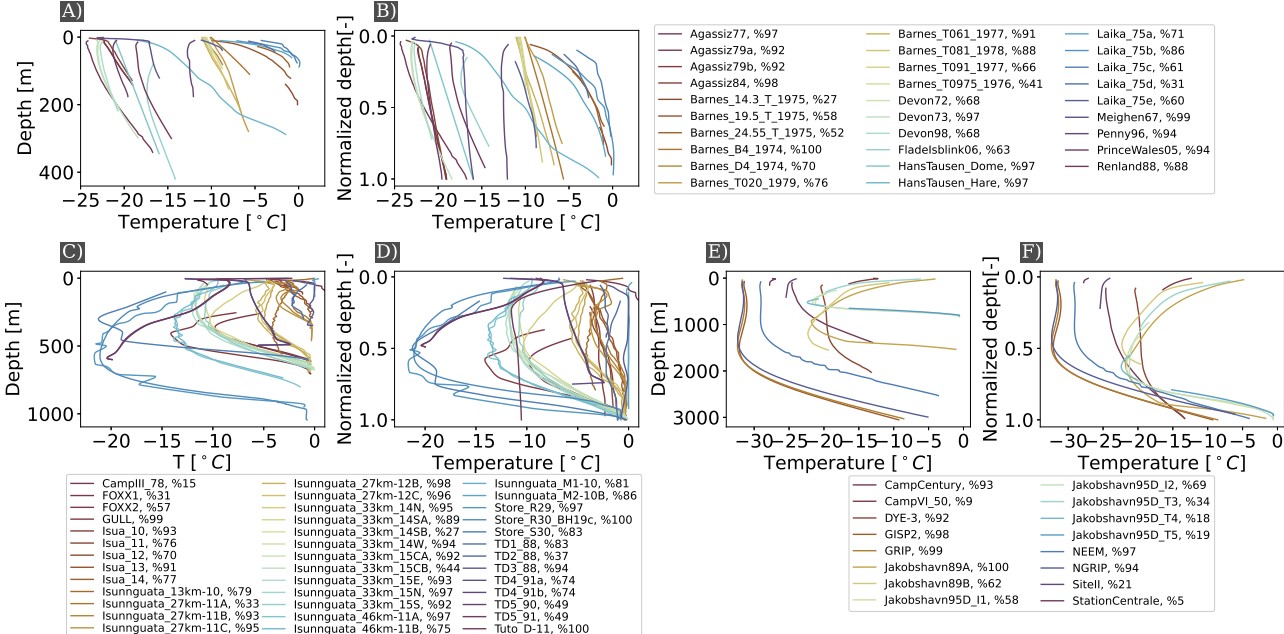

**Figure 3.** Overview of all ice temperature profiles in the database expressed in both absolute depth (**A**),(**C**),(**E**) and normalized depth (**B**),(**D**),(**F**). For visibility, the profiles are divided into local ice caps (**A**)-(**B**), marginal ice-sheet sites (**C**)-(**D**), and inland ice-sheet sites (**E**)-(**F**). The local ice thickness coverage of each individual profile is given as a percentage.

## 3 Metadata

The database includes additional supplementary information. For each borehole, this information is stored in the file "meta.csv", which contains a total of 18 metadata fields (Table 2). Every measurement entry in the database is labeled with a unique **borehole ID**, and a non-unique alternative more descriptive **place name**. This alternative place name was deemed useful as, in some cases, ice temperature profiles have been measured over different campaigns by different people who have used different nomenclature for the same sites. Furthermore, borehole ID's are generally defined by either the group carrying out the mea-



surements or by the site of the measurement. These borehole IDs are somewhat inconsistent, however, based on the differing conventions used by previously published studies.

Whenever specific locations are available, that of each ice temperature borehole is provided in degrees east **longitude** and degrees north **latitude**. In some cases, locations have been shown in figures in accompanying literature, from which approximate locations were georeferenced. In addition to the position of the borehole, three further sets of information are

provided relating to geographic setting. **Site type** describes whether the profile samples the Greenland Ice Sheet or a local ice cap. **Geographic location**, which provides a more descriptive general location of borehole sites using recognized regional geographic names. Lastly, the **source** of the **location** information.

The local **ice thickness** at the borehole location is provided in meters, along with the **ice thickness source**. When the original literature does not report local ice thickness, an estimate is provided using BedMachine v3 (Morlighem et al., 2017). In these

instances, the ice thickness is useful to determine the fraction of the ice depth, from the surface to the bed, covered by the given temperature profile, given as the percentage **coverage** in another field. Also listed are the depth in meters from the surface of the ice sheet to the uppermost temperature measurement of a given profile (**depth of top**), and the lowermost temperature measurement (**depth of bottom**).

For each profile, **data source**, describes the source of the temperature data, including links to open-access data repositories

used by authors to share the original data, and the **science source**, which lists any original scientific publication that describes or mentions the temperature data. The related **DOI** for both the data and science source categories are also reported when available. **Date** describes the common era year when the temperature profile was measured. In the best cases, this includes the month and day a given profile was measured. Finally, a field for additional **notes** is included, presenting any further information (relating to, for example, temperature, thickness or location) that does not fit into the above categories.

Despite the best efforts of our community, there are some missing metadata fields in the database, denoted by a question mark. The database is being continually updated. Any additional information, such as missing, updated or new temperature data, may be added to the database by contacting the author team here, or opening a GitHub issue at https://github.com/GEUS-Glaciology-and-Climate/greenland_ice_borehole_temperature_profiles/issues (Last Access 1 March 2022). We use GitHub issues to resolve and identify and clarify problems in the ice temperature datasets, and we also welcome any new

or missing data into the database via GitHub issues.



**Table 2.** Description of the database metadata structure. The fields are bolded where they appear in the text.

| Field | Description |
| --- | --- |
| Borehole ID (text) | Unique name for each measured profile. |
| Place name (text) | Non-unique name for the site at which the profile is located. |
| Geographic location (text) | General region within which the site is located. |
| Site type (text) | Whether the profile samples the 'ice sheet' or a 'local ice cap', |
| Date (YYYY-MM-DD) | Date of the temperature profile measurement. Nearest year if specific date not available. |
| Longitude (°E) | Longitude of the borehole. |
| Latitude (°N) | Latitude of the borehole. |
| Location source (text) | Source of the borehole coordinates. |
| Ice thickness (m) | Ice thickness measured or estimated at the borehole. |
| Ice thickness source (text) | Source of the ice thickness, citation information and link to original open-access data repository used. |
| Depth of top measurement (m) | Depth to the uppermost temperature measurement in the borehole. |
| Depth of bottom measurement (m) | Depth to the lowermost temperature measurement in the borehole. |
| Coverage (%) | Fraction of the ice depth, from the surface to bed, covered by the ice temperature measurements. |
| Data source (text) | Source of the temperature data. |
| Data DOI (text) | DOI of the temperature data (if available). |
| Science source (text) | Scientific publication about the temperature data (if available). |
| Science DOI (text) | DOI of the science reference (if available). |
| General note (text) | General notes associated with any given temperature profile. |

## 4   Data Summary

Of the 85 deep ice temperature profiles making up the database, 60 originate from Greenland and 25 from the Canadian Arctic Figure 1(Figure 1). Of the 60 profiles from Greenland, 56 are from the ice sheet, and four are from peripheral glaciers and ice caps. All 29 peripheral glaciers and ice cap temperature profiles are plotted together in Figure 3A, B. The peripheral glacier and ice cap profiles are relatively shallow: the deepest reaches 420 m at a site where ice thickness is estimated to be 520 m. Profiles from the ice sheet are divided into 40 marginal profiles (Figure 3C, D) and 16 inland profiles (Figure 3E, F), according




to proximity to the main flow divide. The profile at GISP2, reaching bedrock at depth 3053 m and with a local ice thickness coverage of 98%, is the deepest measurement in the database. Overall, 9 of the 85 profiles reach depths >1000 m.

The inland profiles also exhibit the coldest temperatures, with temperatures reaching a minimum of around -32°C. Thirty-four of the database's 85 profiles are from locations characterized as "cold-bedded", where the ice appears to be frozen to the bedrock. This characterization is assigned based on the observed bottom values of the profile. At some sites we cannot speculate on the basal thermal state based on temperature observation alone. Eight of these "cold-bedded" profiles extend to >95% of local ice thickness (i.e., database coverage >95%). This contrasts with 44 "warm-bedded" sites where the ice at the bed is at pressure melting point. Ten of these warm-bedded profiles extend to >95% of local ice thickness. The basal thermal state of the three sites CampIII_78, CampVI_50, and SiteII, cannot be characterized, as these boreholes covered less than 25% of the local ice thickness. Overall, 38 of the 85 profiles have a depth coverage >90%.

The temporal distribution of the profiles shows how the number of available ice temperature measurements has increased over time (Figure 4). The temperature profiles span a period of 70 years. This implies that the shallow measurement points cannot necessarily be expected to reflect current ice temperatures. Arctic air temperatures have warmed markedly during the past seven decades. The response time for deep ice temperature variations, however, is usually much longer than 70 years (Cuffey and Paterson, 2010). For example, at a low-accumulation site in the ice-sheet interior, a surface air temperature signal from c. 50 years ago can likely influence present-day ice temperatures at >50 m below the ice-sheet surface (Muto et al., 2011; Karlsson et al., 2020).

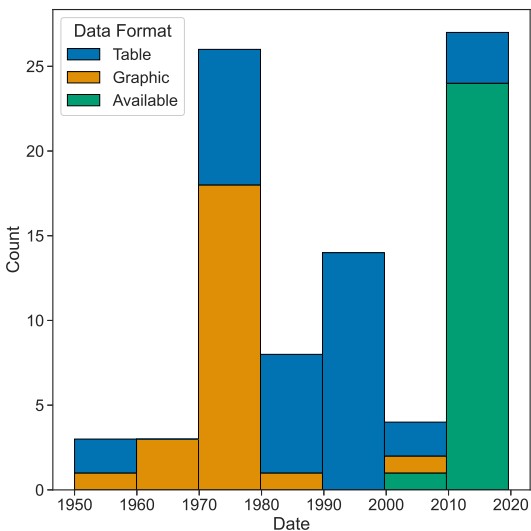

**Figure 4.** Histogram showing the temporal distribution of temperature profile measurements, color coded based on the format the original data was available as. The profiles entered into the database were available as: digitized graphic (orange), digitized table (blue), or is already published and digitally available (green).





The majority of the ice temperature profiles presented here have not been previously available in digital format. Profiles
from only 25 boreholes (c. 32% of the database) have been previously published in open-access data repositories (Ryser et al.,
2014; Harper, 2017; Doyle et al., 2018a; Hubbard et al., 2021b; Law et al., 2021). The public availability of these profiles
highlights the laudable open-science trend in the natural sciences. While we present derived products of these profiles, we
provide links to the open-access repositories where data were originally deposited and encourage database users to recognize
and cite these previous data repository sources where applicable. Profiles from 54 boreholes (c. 68% of the database) are
made digitally available here for the first time. This includes profiles from 24 boreholes that were provided as tabulated data
by community members, profiles from 24 boreholes that were digitized from previously published graphics, and 6 that were
previously available as published tables.

## 5   Sources of Uncertainty

Borehole ice temperature measurements are associated with at least three main sources of uncertainty: measurement uncer-
tainty, depth uncertainty and disturbance uncertainty. There are other sources of uncertainty that we do not consider here,
such as self-heating of the sensor, sensor calibration and leakage path to name a few (e.g. Clow et al., 1996; Clow, 2008).
Measurement uncertainty reflects the accuracy with which a temperature sensor can measure ice temperature. This uncertainty
is influenced by both the design and calibration of the temperature sensor. It is reasonable to suggest that temperature sensor
accuracy has improved by at least an order of magnitude from 1950 to today, from perhaps ±0.1 °C in the 1950s to an effective
limit of better than ±0.01 °C today. This trend primarily reflects improvements in the down-borehole technology deployed in
glaciological applications, not the absolute accuracy of temperature sensors available for public purchase.

The issue of depth uncertainty is greatest in ice temperature profiles measured by manually-deployed loggers. Manual
loggers can have a characteristic depth uncertainty of ±1%. By contrast, automated winch-deployed loggers typically use
rotary encoders to measure logger depth with an uncertainty better than ±0.01%. For example the GISP2 borehole has a drill
depth of 3053 m but a logger depth of only 3050 m, which represents a discrepancy of ±0.001%(Cuffey et al., 1995). At deep
interior sites, the temperature gradient in the lower part of the ice column is typically of the order of 0.02 K/m (cf. Figure
3E,F). At fast moving sites, the temperature gradient may be even larger (cf. Figure 3C,D) thus increasing the temperature
uncertainty associated with depth uncertainty. At shallow sites, temperature gradients are typically small (cf. Figure 3A,B)
making the depth uncertainty negligible. For example, at Camp Century the temperature profile was likely measured with a
depth accuracy higher than ±0.1%. With a measured basal temperature gradient of 17.5 °C/km, this is equivalent to an ice
temperature uncertainty of ±0.025 °C at the maximum borehole depth of 1388 m. There are, however, some site settings –
especially when using manually-deployed loggers, logging non-vertical boreholes, or measuring extreme basal temperature
gradients – in which depth uncertainty can actually play a larger role in ice temperature uncertainty than temperature sensor
accuracy.

Disturbance uncertainty, which is associated with ice temperatures equilibrating after drilling, is a third source of measure-
ment uncertainty. In boreholes filled with drilling fluid, disturbance uncertainty can include the development of convection cells



within the equilibrating drilling fluid. Convection cells are expected to occur when the temperature gradient in the ice exceeds a critical value dependent on the sum of a lapse rate term and the critical potential-temperature gradient (Clow, 2014). The size and strength of convection cells is a poorly defined function of drilling fluid density, borehole diameter and ice temperature

gradient. In the Agassiz77 borehole, slight thermal instabilities in the drilling fluid produced temperature offsets of ±0.05 °C around the mean ice temperature gradient. In the NEEM 2011 profile, however, the convection cells between 1500 and 2000 m depth reach c. 50 m vertical length scale and produce ±0.5 °C offsets around the mean ice temperature gradient (Figure 5). The GISP-D temperature log (not presented here) also shows pronounced laminar convection cells between 1600 and 2000 m depth (Clow et al., 1996).

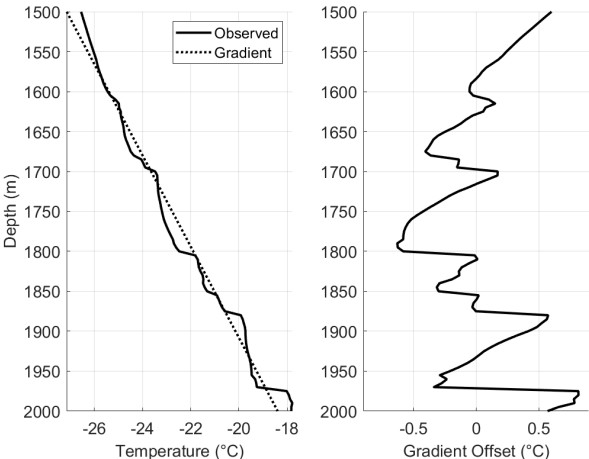

**Figure 5.** Offsets from the mean ice temperature gradient highlight the influence of large (c. 50 m) convection cells forming in the borehole fluid in a deep section of the NEEM ice temperature profile measured in 2011. These borehole fluid convection cells contribute to disturbance uncertainty associated with an equilibrating borehole temperature profile when the borehole logging technique is used. This uncertainty is avoided when sensors or cables are frozen into the ice in other measurement techniques.

Given the ultra-low density of air, air-filled boreholes cannot support analogous convection cells. The disturbance uncertainty associated with convection cells is accordingly negligible in air-filled boreholes. However, temperature disturbances due to changes in atmospheric pressure have been observed in air-filled bore holes in Antarctica. This effect is largest in the top 20 m of the borehole, just below the ice surface, with magnitudes on the order of up to ±0.003 °C (Clow et al., 1996). In water-filled boreholes, such as those created by hot-water drilling, disturbance uncertainty is associated with the sensible heat released

from hot drilling water and the latent heat released by borehole refreezing. This thermal disturbance of hot-water drilling often requires many months to fully dissipate, as the latent energy released by the refreezing borehole warms surrounding ice. Empirical cooling curves of temperature measurements collected multiple times post drilling can usually constrain this thermal disturbance effect to better than ±0.1°C (Humphrey and Echelmeyer, 1990; Ryser et al., 2014; McDowell et al., 2021). Mechanically drilled ice-core holes also experience thermal disturbance, resulting primarily from the sensible heat carried into





the hole by the drill fluid. The associated disturbance uncertainty is, however, much smaller than for hot-water drilled holes
(Clow et al., 2015).

The total measurement uncertainty varies substantially between profiles in this database. Generally, these sources of uncer-
tainty can be viewed as independent from each other, and therefore summed to estimate the total measurement uncertainty for
a given ice temperature profile. Assessing this total measurement uncertainty is most critical for studies focused on absolute

temperature, for example, when re-measuring present day ice temperatures to compare with past ice temperatures at a spe-
cific site. We provide guidance for assessing total measurement uncertainty by contrasting the uncertainty budgets of an older
(Tuto_D-11; 1962) and younger (Store_R30; 2019) ice temperature profile (Table 3). The Tuto_D-11 ice temperature profile
is from an air-filled borehole in which ice temperature was measured with c. 60-year-old sensor technology using a manually-
deployed logger. The Tuto_D-11 data also includes an additional graphical digitization uncertainty described in Section 2. The

Store_R30 ice temperature profile was measured using a fiber optic string frozen into a hot-water-drilled borehole that was
continuously monitored to establish empirical cooling curves. The contrasting uncertainty budgets of these two boreholes (see
Table 3) highlights the value of consulting the original science source publication for site-specific uncertainty assessments.

**Table 3.** Contrasting total measurement uncertainty budgets for the Tuto_D-11 and Store_R30 ice temperature profiles. The conversion of
depth uncertainty into a temperature uncertainty is done assuming a characteristic temperature geothermal gradient of 20 °C/km.

| Uncertainty Source | Tuto_D-11 (1962) | Store_R30 (2019) |
|---|---|---|
| Measurement | ±0.1 °C | ±0.01 °C |
| Depth | ±0.01 °C | negligible |
| Disturbance | negligible | ±0.05 °C |
| Digitization | ±0.1 °C | negligible |
| **Total Uncertainty** | **±0.21 °C** | **±0.06 °C** |

## 6  Comparison with Simulated Ice Temperatures

Numerical ice flow models are critical tools for projecting the future sea-level rise contribution from the Greenland ice sheet.

While the simulated initial states are typically evaluated against observed ice-sheet form (i.e. thickness) and flow (i.e. velocity),
the relative paucity of ice temperature data means that they are typically not evaluated against observed ice-sheet thermal state
(Goelzer et al., 2020). Given the sensitivity of ice viscosity to ice temperature, biases in the thermal state of ice-sheet simula-
tions likely contribute to biases between simulated and observed recent Greenland ice loss (Goelzer et al., 2017; Aschwanden
et al., 2019; Law et al., 2021). Identifying and addressing these biases in thermal state is critical for improving projections of

Greenland ice loss (Aschwanden et al., 2021). To help shape best practice for using the database that we present, we describe an
approach for evaluating a contemporary ice-sheet simulation against all observed Greenland deep ice temperature observations.



Several process-level studies of potential heat sources have been performed, which compare individual observed temperature profiles from local areas with temperature profiles modeled by a thermal, or themo-mechanical, ice flow model (Iken et al., 1993; Lüthi et al., 2002; Harrington et al., 2015; Lüthi et al., 2015; Meierbachtol et al., 2015; McDowell et al., 2021; Law et al., 2021; Maguire et al., 2021). Although these studies featured different local areas, the comparisons generally showed that models tend to underestimate englacial temperatures, and thus need to incorporate additional heat sources in order to reproduce observed ice temperature profiles. Suggested additional heat sources include cryo-hydrological warming, which transfers latent heat when surface melt water flows through englacial pathways and re-freezes, as well as deformational heating and basal water heat flux (Funk et al., 1994; Wohlleben et al., 2009; Phillips et al., 2013; Lüthi et al., 2015; Zekollari et al., 2017; Karlsson et al., 2020).

The contemporary thermomechanical ice-sheet simulation we adopt is the Parallel Ice Sheet Model (PISM; Bueler and Brown (2009)) simulation of Aschwanden et al. (2016). This simulation represents the form, flow and thermal state of the Greenland ice sheet in c. 1990, following a 125 ky paleo-climatic spin-up, followed by another 2 ky of transient equilibrium with mass-flux adjustment forcing to minimize misfit against observed ice-sheet thickness and extent. This ice-sheet simulation has a horizontal resolution of 900 m and a vertical resolution of 20 m. PISM uses an enthalpy scheme for the conservation of energy calculation, in order to accommodate heat transfer in both freezing and temperate ice (Aschwanden et al., 2012). While the spin-up is meant to approximate the Greenland Ice Sheet at a specific time slice (c. 1990), we are comparing this simulated thermal state with temperature profiles observed over a 70-year time span. Shallow profiles located close to the margin can experience significant changes in ice thickness and temperature on this time scale.

At locations of observed temperature profiles, modeled vertical temperature profiles were extracted from the PISM simulation based on the nearest neighbor grid point. Only temperature profiles from the Greenland Ice Sheet were included in this analysis. The PISM vertical temperature profiles were transformed from height above bed, to normalized depth below surface, and linearly interpolated to the vertical resolution of the normalized depth field of the database. Further, at a few drill site locations, the modeled ice was too thin, and did not have enough vertical grid points to interpolate the modeled temperatures to a proper normalized depth axis. It was therefore necessary to exclude profiles from the analysis where the simulated ice thickness was less than 5 vertical grid points (50 m). Additionally, in cases of temporally-repeated observed profiles, only the most recent measurement was included in this analysis see Table 4.

Out of the 85 temperature profiles from the temperature database, 56 were ultimately suitable for inclusion in this comparative analysis of observed and modeled ice temperatures. The observed temperature profiles were divided based on three characteristic regimes. First, whether profiles were located in the accumulation area (16 profiles) or ablation area (38 profiles). Secondly, whether the basal thermal state of profiles were considered warm-bedded (44 profiles) or cold-bedded (9 profiles). Finally, whether profiles were located in high-strain regions (10 profiles) or low-strain regions (46 profiles). The strain rate characterization was subjectively assigned by the author team, where high-strain rate was defined for sites approaching channelized glacier flow. Figure 6 shows the observed temperature profiles for each of the three regimes.



**Table 4.** Contrasting total measurement uncertainty budgets for the Tuto_D-11 and Store_R30 ice temperature profiles. The conversion of depth uncertainty into a temperature uncertainty is done assuming a characteristic temperature geothermal gradient of 20 °C/km.

|  | **Surface mass balance regime** | **Basal thermal state regime** | **Ice dynamic regime** |
|---|---|---|---|
| Initial Number of profiles | 16 accumulation<br>38 ablation<br>2 unknown | 9 cold bedded<br>44 warm bedded<br>3 unknown | 10 high strain rate<br>46 low strain rate |
| Total | 56 | 56 | 56 |
| Number of profiles after exclusions | 16 accumulation<br>31 ablation | 5 cold bedded<br>40 warm bedded | 10 high strain rate<br>38 low strain rate |
| Total | 47 | 45 | 48 |

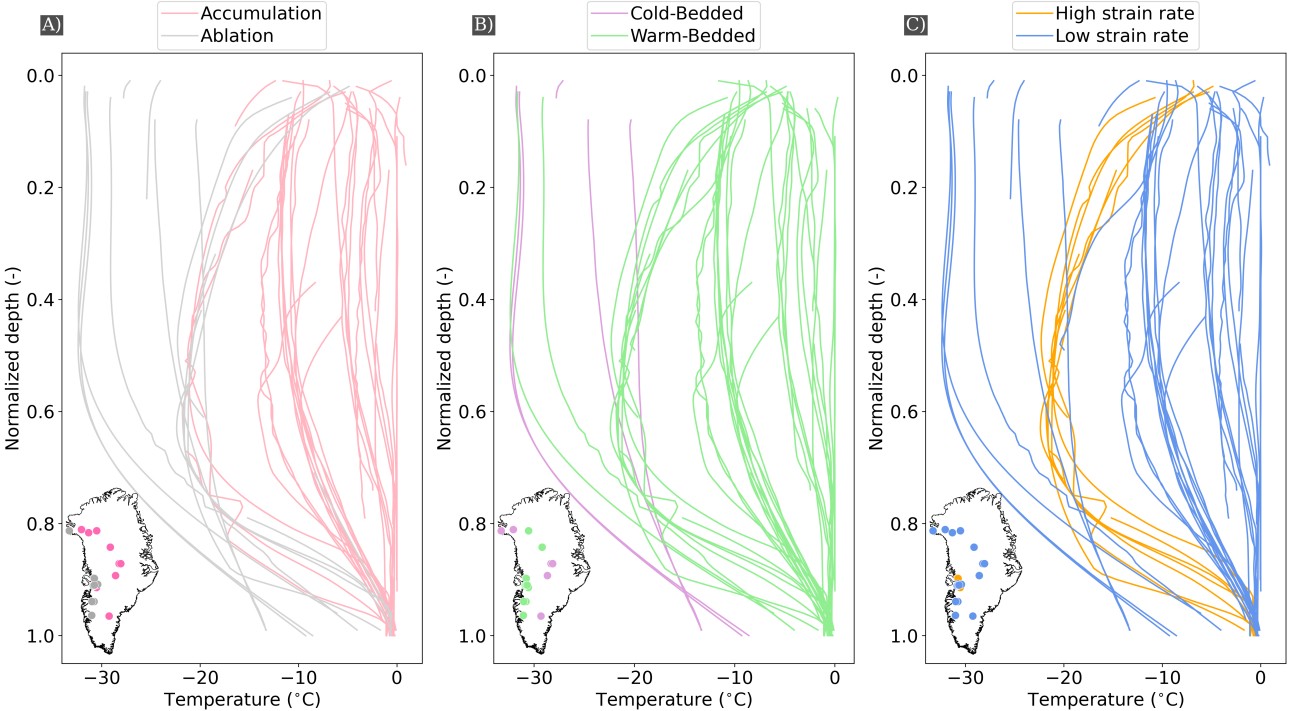

**Figure 6.** Observed temperature profiles on normalized depth scale for three characteristic regimes. **(A)**: Surface mass balance regime with profiles located in the ablation area (pink) or the accumulation area (gray). **(B)**: Basal thermal state regime with profiles either cold-bedded (purple) or warm-bedded (green). **(C)**: Ice dynamic regime with profiles located either in high-strain rate regions (orange) or low-strain rate regions (blue).

The difference between the modeled and observed profiles was calculated at 0.01 normalized vertical resolution and then aggregated into coarser vertical layers at 0.1 normalized vertical resolution (Figure 7). This method of examining profiles in vertical sections provides better insight on depth-variations of model bias than reducing every profile into a single metric of best fit. This vertical boxplot analysis can also provide some insight on which terms in the energy balance – advection, diffu-

sion, heat production, or latent heat transfer – may be responsible for differences between modeled and the observed profiles, as the relative importance of these terms varies with depth. Where our comparison boxplots show negative temperature differences, modeled temperature values are smaller than observed temperatures, implying that this PISM run is underestimating temperatures. Vice versa where boxplot temperature differences are larger than 0 °C.

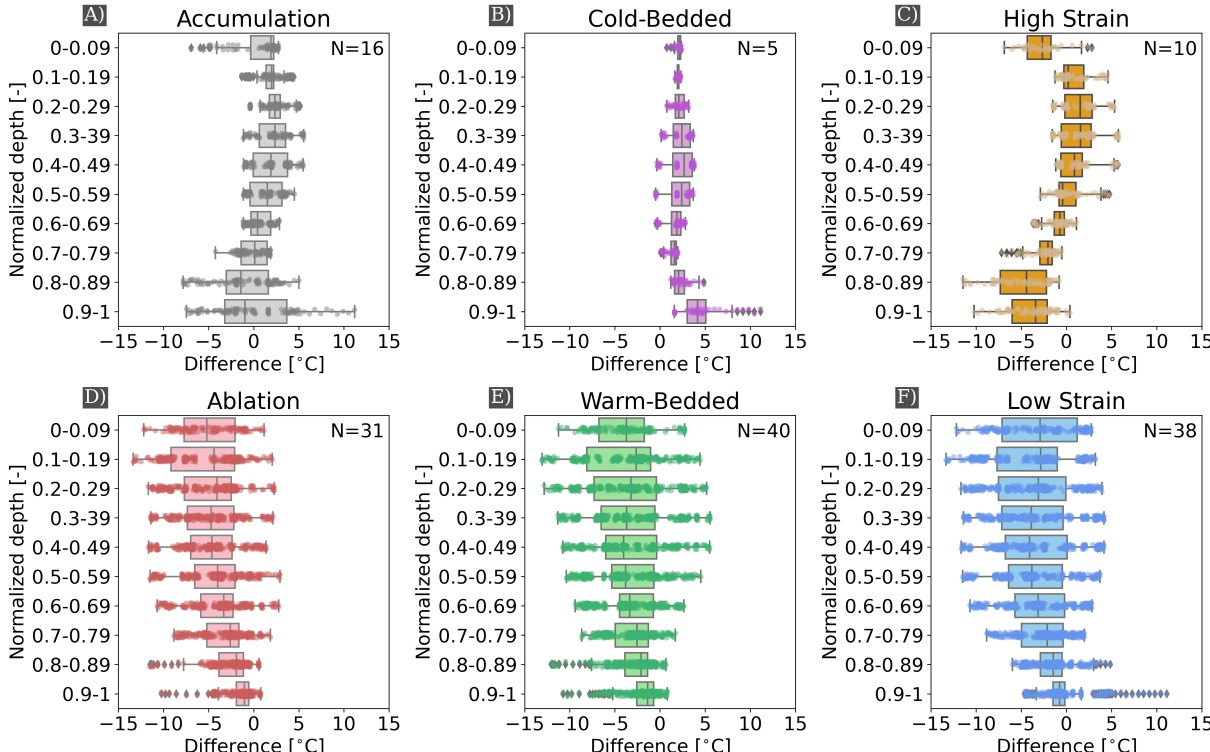

**Figure 7.** Each plot shows the difference between the modeled minus observed normalized depth profiles aggregated into coarser vertical layers of 0.1 resolution. **(A), (D)**: Surface mass balance regime. **(B), (E)**: Basal thermal state regime. **(C), (F)**: Ice dynamic strain regime. The N value shows the number of profiles the boxplot is based on. Points plotted on top of each boxplot shows individual temperature difference values for all of the profiles at that depth within that category.

The surface mass balance regime comparison generally highlights a better PISM fit to observations in the accumulation

area than the ablation zone (Figure 7). This is generally expected, since the accumulation area more closely represents the theoretical steady-state temperature profiles upon which classic ice flow heat equations are predicated. The accumulation boxplot shows that the mean difference value shifts from being positive in the upper part of the profile to negative, with a





greater spread in differences, in the bottom 20% of the profile. In comparison, both the ablation and temperate boxplots, which share strong overlap in comprising profiles, are clearly less well captured by PISM. These warmer ablation area profiles show a

larger spread in difference values throughout the entire vertical profile. Importantly, both boxplots show a systematic negative mean difference at all depths, meaning the simulation consistently underestimates ice temperatures. This may suggest that one or more heat sources are being poorly represented in warmer ablation areas in this particular PISM simulation. This is a general problem for solving the heat equation in all ice flow models. Previous studies similarly showed poor representation of the ablation area and the need to include additional heating sources to reproduce the higher observed englacial temperatures

(McDowell et al., 2021).

After excluding profiles where modeled thickness is less than 50 m, the cold-bedded regime boxplot represents the smallest sample size of just five profiles (Camp Century, Dye-3, GISP2, GRIP, and Station Central). These profiles are all drilled at inland locations where the ice is moving relatively slowly. This generally limits the role of horizontal heat advection at these sites. In contrast to the warm-bedded and ablation boxplots, the cold-bedded boxplot shows a systematic positive mean

difference over all vertical layers, meaning that modeled temperatures generally overestimate observed temperatures over the entire column of ice. The largest differences between modeled and observed temperatures (+4 °C on average) however, are found near the ice-sheet bed. Here, where vertical ice advection becomes small, diffusion of geothermal flux should be the most influential term of the heat equation. As heat diffusion is generally well-represented and well-constrained in all ice flow models, this may suggest that this particular PISM simulation employs too high a geothermal heat flux into the ice-sheet base.

We note that the geothermal flux is poorly constrained under the Greenland Ice Sheet (Colgan et al., 2022) and over long spin-up periods, even small deviations in geothermal heat flux can result in large variations in ice temperature.

The shape of temperature profiles in fast-flowing high-strain regions is expected to have a C-shape associated with strong horizontal advection of upstream ice temperatures (Cuffey and Paterson, 2010; Iken et al., 1993). While this advection of relatively cold upstream ice cools the middle depths of the ice temperature profile, in comparison to the surface boundary

condition, deformational heating warms the lower depths (Cuffey and Paterson, 2010). The high strain boxplot shows that the PISM simulation appears to systematically underestimate ice temperatures in the lowermost 50% of the ice sheet. This bias reaches up to -5 °C, on average, between 80-90% depth. This underestimation of ice temperatures near the bed may result from this particular PISM simulation underestimating strain heating associated with enhanced ice deformation in softer Last Glacial Period ice (e.g. Lüthi et al., 2002; Karlsson et al., 2013). Generally, however, it is challenging to simulate temperature profiles

in high-strain ice flow regimes, as this requires correct parameterization of both local and upstream heat budgets, as well as the horizontal advection profile linking these heat budgets, in order to correctly represent steep gradients in the ice temperature profile (Cuffey and Paterson, 2010).

The low strain boxplot shows a systematic negative mean difference value across all ice depths. This suggests that, in slow-flowing regions, the model generally underestimates ice temperatures. This might be explained by this particular PISM

simulation employing a spin-up surface boundary condition that was either too cold air temperature or too high accumulation rate. As vertical ice advection is dependent on accumulation rate, overestimated accumulation rates can result in overestimated vertical advection rates, which enhance the advection of relatively cold near-surface ice deeper into the ice sheet. The low strain



regime profiles are diverse, however, comprising both ablation and accumulation area profiles – and with both cold and warm beds. The low strain regime generally has the largest spread of difference values. This makes it difficult, in comparison to the
other boxplots presented here, to interpret a process-level mechanism for the systematic cold bias across these diverse sites.

## 7 Outlook

We have compiled 85 ice temperature profiles from 79 individual boreholes from Greenland and the Canadian Arctic into a comprehensive and consistent community resource. This constitutes an increase of +54 ice temperature profiles over what was previously available in open-access data repositories (68% of the database). The scientific community is encouraged to
contribute missing or new data or metadata to the database either by contacting the author team or using the GitHub issue pathway. While we expect the primary use of this database to be evaluating simulated 3-D ice temperature fields from thermo-mechanical ice flow models, there may soon be a sufficient number of ice temperature profiles from the Greenland Ice Sheet and peripheral ice caps to use machine learning approaches to spatially interpolate ice temperatures between boreholes on the basis of independent geophysical variables. This database may possibly serve as a modest first step towards addressing this,
and other, emerging challenges associated with understanding the thermal state of ice in Greenland and the Canadian Arctic.

We provide an initial comparison between observed ice temperature profiles and those simulated by the PISM ice flow model. This comparison highlights the inherent challenge of comparing 1-D observational datasets with uneven spatial distribution to a simulated 3-D temperature field. These challenges include: the uneven ice thickness between modeled and observed geometry, especially at marginal sites; resolving the difference between the systematic biases and random error in the modeled tempera-
ture profiles; and confronting fundamental uncertainty in basal thermal state due to complex processes. In this comparison, the poor representation of cryo-hydrological warming, or the transfer of latent energy, appears to be contributing to a model cold bias at peripheral ice-sheet sites. Similarly, poor representation of deformational heating and vertical ice stratigraphy appears to be contributing to a model cold bias at high-strain rate sites. Clearly, thermomechanical models provide the most reliable solution to the heat equation in low-strain, cold-bedded inland areas.

*Data availability.* The Greenland deep ice temperature database described here is available at https://doi.org/10.22008/FK2/3BVF9V(Mankoff et al., 2022). The source or inputs to the database are available at https://github.com/GEUS-PROMICE/greenland_ice_borehole_temperature_ profiles(Last Access 3 March 2022; Mankoff (2022)). The supplementary metadata described in this article, including profile classification is available in the supplementary information of this article.

Users of this product should cite the data repository (Mankoff et al., 2022), and this descriptive data article. Users of this product should
also cite the original "Data Source" and "Data DOI" repositories when they are available. We encourage data users to similarly consult relevant "Science Source" articles, which can be found in the metadata for each profile. Furthermore, data users who are generating derivative products from database entries are requested to invite collaboration with the "Science Source" team associated with the database entries being utilized.





*Author contributions.* Author contribution is captured following the CRediT system. Conceptualization by WC, KM. Data curation by WC,
KM, HT, GDC, DF, CZ, MPL, BV, DDJ, IM, HZ, TM, SHD, AA, B. Hills, RL, JH, NH, B. Hubbard, PC, MJ. Formal analysis by AL, SHD,
AA, RL, JH. Funding acquisition by JH, B. Hubbard, PC. Investigation by AL, SHD, AA, RL, JH, B. Hubbard, PC. Interpretation by AL,
AS, HT, GDC, MPL, NBK, HZ, SHD, RL, JH, B. Hubbard, PC. Methodology by AL, KM, WC, GDC, NBK, RL, JH, B. Hubbard, PC.
Project administration by KM, WC, RSF, SAK, JH, B. Hubbard, PC. Resources by RSF, SAK. Software by KM. Supervision by KM, WC,
JH, B. Hubbard, PC. Validation by AL, KM, WC, AS, NBK, SHD. Visualization by AL, KM. Writing pre-submission by AL, KM, WC,
GDC, NBK, HZ, SHD, RL, B. Hubbard, PC.

*Competing interests.* The authors declare that they have no conflict of interest.

*Acknowledgements.* Development of this database by the Geological Survey of Denmark and Greenland (GEUS) was supported by the Pro-
gramme for Monitoring of the Greenland ice sheet (www.promice.dk), Independent Research Fund Denmark (IRFD) award 8049-00003B,
and Villum Foundation award 00022885. Ice coring on Canadian Arctic ice caps was primarily supported by the Polar Continental Shelf
Project and the Geological Survey of Canada (Natural Resources Canada), with additional funding from the Canadian Foundation for Cli-
mate and Atmospheric Science (Prince of Wales Icefield 2005) and Japan's Science and Technology Agency (Penny ice cap, 1996). Harry
Zekollari acknowledges funding from the Fonds de la Recherche Scientifique – FNRS (postdoctoral grant – chargé de recherches) and PRO-
TECT, which has received funding from the European Union's Horizon 2020 research and innovation programme under grant agreement
No. 869304. Ice temperature measurements on Store Glacier were supported by the European Union's Horizon 2020 research and inno-
vation programme under grant agreement 683048 (RESPONDER), the National Environment Research Council (NERC) (SAFIRE grants
NE/K006126 and NE/K005871/1), and an Aberystwyth University Capital Equipment grant. A. Aschwanden was supported by NASA grant
20-CRYO2020-0052. R. Law acknowledges funding from NERC Doctoral Training Partnership studentship grant NE/L002507/1. Mylène
Jacquemart acknowledges funding from the Swiss National Science Foundation (grant SNF 184634 PROGGRES). Harper, Humphrey, Meier-
bachtol, McDowell and Hills funded by the U.S. National Science Foundation, Office of Polar Programs (grant nos. 1203451 and 0909495).
G. Clow acknowledges funding from the U.S. Geological Survey and the U.S. National Science Foundation, Office of Polar Programs. Martin
P. Lüthi acknowledges funding from the Swiss National Science Foundation, project number 200021_127197.

We also thank Baptiste Vandercrux and Joseph A. MacGregor for their discussion during the development of this database.



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
