# Peer review of "Greenland and Canadian Arctic ice temperature profiles database"

_The Cryosphere, 2022_

## Referee Comment (RC1)

**Comments on tc-2022-138**

In the manuscript, 85 ice temperature profiles from Greenland and Canadian Arctic are collected and compiled into a coherent database, of which 68% are digitally available for the first time. In addition, 56 observed temperature profiles from Greenland ice sheet are compared with the PISM simulation of Aschwanden et al. (2016) to identify the possible heat source for misfit of observed and modeled temperature profile. The established database provides a good date source for science community to understand the thermal state of Greenland ice sheet and local ice caps in the Canadian Arctic. Generally, the manuscript is well written and can be accepted after addressing the following questions.

(1) In Table 1, four measurement methods are presented. However, the digital sensor string and thermistor string are not mentioned in the text. It is better to explain more details of the two measurement methods.

(2) In Figure 1, the drill site location in the green box is not shown in Figure 1A. It is better to show the Jakobshavn glacier.

(3) Line 95: Please check the ice thickness in Tuto_D-11 borehole, it looks from the Figure 2 that the ice sheet thickness is 200 ft, which is about 61 m.

(4) In the database, it is better to presented the temperature measurement methods (e.g., type and accuracy of temperature sensors) and depth measurement methods (e.g., type and accuracy of encoder) for the readers to evaluate the uncertainty of data source.

(5) Line 210-220: the paper of V. Zagorodnov et al. presented more detailed disturbance uncertainty of mechanical drill and some discussion can be included in the manuscript. (Zagorodnov, V., Nagornov, O., Scambos, T. A., Muto, A., Mosley-Thompson, E., Pettit, E. C., & Tyuflin, S. (2012). Borehole temperatures reveal details of 20th century warming at Bruce Plateau, Antarctic Peninsula. The Cryosphere, 6(3), 675-686.)

(6) Section 6: Please provide more details how the author determined surface mass balance regime, the basal thermal state regime and ice dynamic regime. A table with accumulation/ ablation rate, basal temperature and strain rate is preferred.

In addition, some technical errors should be corrected.

(1) Line 15: "the thermal state of the sheet" should be "the thermal state of the ice sheet".

(2) Line 20: "thermo-mechanical" or "thermodynamic" or "thermomechanical"? Descriptions should be consistent throughout the manuscript.

(3) Line 25: Please check the sentence "borehole logging where a temperature sensor is moved up or down the borehole measuring either "continuously" as the probe moves down". Borehole logger is used only when moves down? or, it can be used when moving down or up.

(4) Line 25: "fiber-optic distributed temperature sensing", "Fiber optic distributed sensing string"? The hyphenation between fiber and optic should be consistent throughout the manuscript.

(5) Figure 1: The units of Celsius should have the same format throughout the paper.

(6) Section 4: There are two "Figure 1" in the first sentence of the section.

(7) Line220 and 230: "hot-water-drilled borehole" or "hot-water drilled borehole"? The style should be consistent throughout the manuscript. I think it should be "hot-water drilled borehole".

(8) Table 4: The caption of the table 4 is the same as the table 3.

(9) The style of the references should be consistent, for example, the first letter of each word in the title of references should be lowercase. Please carefully check your references.

---

## Referee Comment (RC2)

**Greenland and Canadian Arctic ice temperature profiles**

Anja Løkkegaard[1,2], Kenneth Mankoff[1], Christian Zdanowicz[3], Gary D. Clow[4], Martin P. Lüthi[5],
Samuel Doyle[6], Henrik Thomsen[1], David Fisher[7], Joel Harper[8], Andy Aschwanden[9], Bo M. Vinther[10],
Dorthe Dahl-Jensen[10], Harry Zekollari[11], Toby Meierbachtol[8], Ian McDowell[12], Neil Humphrey[13],
Anne Solgaard[1], Nanna B. Karlsson[1], Shfaqat Abbas Khan[2], Benjamin Hills[14], Robert Law[15],
Bryn Hubbard[6], Poul Christoffersen[15], Mylène Jacquemart[16], Robert S. Fausto[1], and William T. Colgan[1]

[1]Geological Survey of Denmark and Greenland, Copenhagen, Denmark
[2]DTU Space, Technical University of Denmark, Denmark
[3]Department of Earth Sciences, Uppsala University, Uppsala, Sweden
[4]Institute of Arctic and Alpine Research, University of Colorado Boulder, USA
[5]Geographical Institute, University of Zurich, 8052 Zurich, Switzerland
[6]Centre for Glaciology, Department of Geography and Earth Sciences, Aberystwyth University, Aberystwyth, SY23 3DB, UK
[7]Department of Earth and Environmental Sciences, University of Ottawa, Ottawa, Canada, K1N 6N5
[8]Department of Geosciences, University of Montana, Missoula, MT,59812, USA
[9]Arctic Region Supercomputing Center, University of Alaska Fairbanks, Fairbanks, AK, USA
[10]Centre for Ice and Climate, Niels Bohr Institute, University of Copenhagen, Copenhagen, Denmark, 2100
[11]Laboratory of Hydraulics, Hydrology and Glaciology (VAW), ETH Zurich, Zurich, Switzerand; Swiss Federal Institute for Forest, Snow and Landscape Research (WSL), Birmensdorf, Switzerland; Laboratoire de Glaciologie, Université libre de Bruxelles, Brussels, Belgium
[12]Graduate Program of Hydrologic Sciences, University of Nevada, Reno, Reno, NV, USA
[13]Department of Geology and Geophysics, University of Wyoming, Laramie, WY, USA
[14]Department of Earth and Space Sciences, University of Washington, Seattle, WA, USA
[15]Scott Polar Research Institute, University of Cambridge, Cambridge, UK
[16]Laboratory of Hydraulics, Hydrology and Glaciology (VAW), Department of Civil, Environmental and Geomatic Engineering, ETH Zurich, Zurich, Switzerand and Swiss Federal Institute for Forest, Snow and Landscape Research (WSL), Birmensdorf, Switzerland

**Correspondence:** Anja Løkkegaard (aloe@geus.dk)

**Abstract.** Here, we present a compilation of 85 ice temperature profiles from 19 boreholes from the Greenland Ice Sheet and peripheral ice caps, as well as local ice caps in the Canadian Arctic. Only 25 profiles (32%) were previously available in open-access data repositories. The remaining 54 profiles (68%) are being made digitally available here for the first time. These newly available profiles, which are associated with pre-2010 boreholes, have been submitted by community members or digitized from published graphics and/or data tables. All 85 profiles are now made available in both absolute (meters) and normalized (0 to 1 ice thickness) depth scales, and are accompanied by extensive metadata. This metadata includes a transparent description of data provenance. The ice temperature profiles span 70 years, with the earliest profile being from 1950 at Camp VI, West Greenland. To highlight the value of this database in evaluating ice flow simulations, we compare the ice temperature profiles from the Greenland Ice Sheet with an ice flow simulation by the Parallel Ice Sheet Model (PISM). We find a cold bias in modeled near-surface ice temperatures within the ablation area, a warm bias in modeled basal ice temperatures at inland

**Summary of Comments on tc-2022-138.pdf**

**Page: 1**

**Number: 1 Author: brice**  Subject: Highlight  Date: 20/10/2022 21:40:30
In the xlsx supplement 70 boreholes are identify ! but i see 79 on github and dataverse

**Number: 2 Author: brice**  Subject: Inserted Text Date: 20/10/2022 21:40:33
replace with "boreholes data". If you use "profiles", it refers to the "85 ice temperature profiles" and as a result, the percentage should be 29.5%.

**Number: 3 Author: brice**  Subject: Inserted Text Date: 20/10/2022 21:40:36
same here, it should refer to the boreholes and not the profiles

**Number: 4 Author: brice**  Subject: Underline  Date: 20/10/2022 21:23:44
these two values must be included in the xlsx supplement

cold-bedded sites, and an apparent underestimation of deformational heating in high-strain settings. These biases provide process-level insight on simulated ice temperatures.

**1 Introduction**

With the tremendous social implications of sea-level change, the past decade has seen a proliferation of simulations of the
15  current and future geometry and dynamics of the Greenland ice sheet (Bindschadler et al., 2013; Goelzer et al., 2020; Aschwanden et al., 2021). These present-day complex ice flow models build upon the legacy of simpler past thermodynamic models (Letréguilly et al., 1991; Huybrechts et al., 1991; Funk et al., 1994; Calov and Hutter, 1996). Due to the high sensitivity of ice viscosity to ice temperature, the thermal state of the [1]eet is a critical element of these simulations (Colgan et al., 2015). At present, however, the englacial temperature fields of even cutting-edge ice-sheet simulations remain largely unevaluated
20  against observed temperatures (Aschwanden et al., 2019). While recent studies show potential for deriving internal ice temperatures from satellite or airborne data (Macelloni et al., 2019; Jezek et al., 2022), these techniques are not yet widely employed. There are consequently diverse opinions on Greenland's basal thermal state across the current generation of thermo-mechanical ice flow models (MacGregor et al., 2016).

Several different methods for measuring ice temperatures have been used on the Greenland Ice Sheet and in the Canadian
25  Arctic. The methods include: borehole logging where a temperature sensor is moved up or down the borehole measuring either "continuously" as the probe moves down or is stopped to measure at every depth known as "stop-and-go" (Johnsen et al., 1995; Clow, 2008); sensor strings where thermistors are frozen into the ice and ice temperatures are recorded at various depths at the same time (Iken et al., 1993; Ryser et al., 2014); and, fiber-optic distributed temperature sensing (DTS), where the ice temperature is measured near-continuously along the full cable length (Law et al., 2021). We summarize the methods and their
30  advantages and disadvantages in Table 1.

Ice temperature profiles collected in Greenland and the Canadian Arctic have not been systematically compiled into a coherent database. In particular, many pre-1990 ice temperature profiles languished in undigitized reports or gray literature. This presented a clear motivation to assemble ice temperature measurements into a consistent and comprehensive community resource. Here, we describe our compilation of ice temperature profiles from Greenland and the Canadian Arctic into an
35  open-access database with well-documented and uniform metadata for each entry. We include Canadian Arctic ice caps in our predominantly Greenland database, as these regions reside within the domain of some Greenland ice flow models ([2]rasov and Peltier, 2004; Gowan et al., 2021).

The earliest temperature profile in our database is from 1950, when a 125 m profile was measured at Camp VI in West Greenland (Heuberger, 1954). [3]arlier temperature profiles [4]may exist, and incorporating these profiles into the database is an
40  ongoing process. We restrict our database to ice temperature profiles extending well below the depth of the seasonal temperature cycle. At cold, dry sites, this is often approximated to be 10 to 15 m below the surface (Cuffey and Paterson, 2010). Similar to a recent effort to compile surface mass balance observations into a readily accessible common framework (Machguth et al., 2016), we aim to create a community resource that facilitates comparisons between simulated and observed ice temperatures.

Number: 1 Author: brice          Subject: Inserted Text Date: 20/10/2022 21:40:47
ice

Number: 2 Author: brice          Subject: Inserted Text Date: 20/10/2022 21:40:52
e.g.

Author: brice          Subject: Sticky Note   Date: 20/10/2022 21:40:41

Number: 3 Author: brice          Subject: Underline     Date: 20/10/2022 21:24:38
Could you please provide more details:  locations, depth, how many did you find, ... ? Too vague.

Number: 4 Author: brice          Subject: Cross-Out     Date: 10/10/2022 14:44:41

[revised manuscript text omitted]

Number: 1 Author: brice      Subject: Underline      Date: 20/10/2022 21:29:32

Your KML file is in the EPSG 4326 projection. As you use the EPSG 3413 projection for the figures, it would be good to add a Geojson or shp file with these projections in the repository. It's a plus not a must have

Number: 2 Author: brice      Subject: Underline      Date: 20/10/2022 21:32:57

Why did you apply a cubic spline interpolation? Why not simply use a simple piecewise linear interpolation ? The effects of thermodynamical parameters (mainly surface temp, geothermal heat flow, ice velocity, ...) on the ice temperature vary with depth and so a cubic spline is not necessarily the best fit for all depths. Could you explain why you used a cubic spline everywhere ?

Number: 3 Author: brice      Subject: Cross-Out      Date: 10/10/2022 15:28:39

Number: 4 Author: brice      Subject: Inserted Text      Date: 10/10/2022 15:28:33

format

Number: 5 Author: brice      Subject: Inserted Text      Date: 10/10/2022 15:31:53

t

95   to profile. To highlight the potential importance of digitization uncertainty, we propagate the uncertainty associated with a ±2

pixel error in each of the axes defining points for both temperature and depth in the digitized profiles with the thinnest ice

(Tuto_D-11; 48 m) and the thickest ice (DYE-3; 2038 m). The Tuto_D-11 graphic (Davis, 1967)(Figure 2A) is  1,000 pixels

tall and spans 75 m, which yields a ±2 pixel depth uncertainty of ±0.15 m. The Tuto_D-11 graphic is  600 pixels wide and

spans 32 °C, which yields a ±2 pixel temperature uncertainty of ±0.1 °C. In comparison, the DYE-3 graphic (Gundestrup and

100  Hansen, 1984)(Figure 2B) is  1200 pixels tall, spans 2200 m and yields a ±2 pixel depth uncertainty of ±3.6 m. The DYE-3

graphic,  900 pixels wide, spans 8 °C and yields a ±2 pixel temperature uncertainty of ±0.02 °C. These end-member scenarios

highlight how digitization uncertainty varies from graphic to graphic.

[Figure]

**Figure 2.** (**A**): Graphic from which the Tuto_D-11 profile is digitized (Davis, 1967). This figure is reproduced from a US Government pub-
lication. (**B**): Graphic from which the DYE-3 profile is digitized (Gundestrup and Hansen, 1984). This figure is reproduced with permission
from the International Glaciology Society.

All ice temperature profiles – from both digitized and tabulated sources – were standardized by interpolating between

points, using cubic spline interpolation with no overshoot, in order to resolve the two common depth scales: the absolute

105  depth scale at 1 m vertical resolution (Figure 3A, C, E) and the normalized depth scale at non-dimensional 0.01 vertical

resolution (Figure 3B, D, F). The latter allows easy comparison between sites of different ice thicknesses, and is useful in

overcoming slight differences in ice thickness when comparing observed and modeled temperature profiles. The normalized depth scale temperature file is named "temperature_dnorm.csv". The absolute depth scale temperature file, which is expressed as temperature with depth from the surface, is named "temperature.csv". In each file, 'NaN' means that no temperature data
110 is available at the given depth, and in the absolute depth file '-999' refers to an elevation below bedrock. Ice thicknesses vary significantly across the database profiles, from 3085 m at NGRIP to 48 m at Tuto_D-11, so the number of -999 below-bed null-values varies between sites. This is because the temperature vector length is constant across the database entries, but the ice thickness varies across database entries.

[Figure]

**Figure 3.** Overview of all ice temperature profiles in the database expressed in both absolute depth (**A**),(**C**),(**E**) and normalized depth (**B**),(**D**),(**F**). For visibility, the profiles are divided into local ice caps (**A**)-(**B**), marginal ice-sheet sites (**C**)-(**D**), and inland ice-sheet sites (**E**)-(**F**). The local ice thickness coverage of each individual profile is given as a percentage.

**3 Metadata**

115 The database includes additional supplementary information. For each borehole, this information is stored in the file "meta.csv", which contains a total of 18 metadata fields (Table 2). Every measurement entry in the database is labeled with a unique **borehole ID**, and a non-unique alternative more descriptive **place name**. This alternative place name was deemed useful as, in some cases, ice temperature profiles have been measured over different campaigns by different people who have used different nomenclature for the same sites. Furthermore, borehole ID's are generally defined by either the group carrying out the mea-

Number: 1 Author: brice    Subject: Underline    Date: 20/10/2022 21:35:27
I understand that it is complex to show all the drill site temperature data in one figure but at this scale it is difficult to see which data set is which. In itself, the figure is interesting to show the overall variations in the temperature data but not useful to show specific data sets. Maybe using differing linestyles would help separating out data sets visually. Another solution would be to make a bigger figure as supplementary material.

[revised manuscript text omitted]

Number: 1 Author: brice        Subject: Underline      Date: 20/10/2022 21:37:22
Could you please define what "negligible" is. For some, 0.01 °C is also negligible (e.g. modelers). In other words, what is your defined limit ?

Number: 2 Author: brice        Subject: Underline      Date: 20/10/2022 21:38:23
This paragraph is interesting and shows the importance of providing good constraints for models. You should add a table listing the key parameters used in the model (even-though it is not a paper on PISM) and explain how well they are constrained.

[Figure]

Several process-level studies of potential heat sources have been performed, which compare individual observed temperature profiles from local areas with temperature profiles modeled by a thermal, or themo-mechanical, ice flow model (Iken et al., 1993; Lüthi et al., 2002; Harrington et al., 2015; Lüthi et al., 2015; Meierbachtol et al., 2015; McDowell et al., 2021; Law et al., 2021; Maguire et al., 2021). Although these studies featured different local areas, the comparisons generally showed that models tend to underestimate englacial temperatures, and thus need to incorporate additional heat sources in order to reproduce observed ice temperature profiles. Suggested additional heat sources include cryo-hydrological warming, which transfers latent heat when surface melt water flows through englacial pathways and re-freezes, as well as deformational heating and basal water heat flux (Funk et al., 1994; Wohlleben et al., 2009; Phillips et al., 2013; Lüthi et al., 2015; Zekollari et al., 2017; Karlsson et al., 2020). [1]

The contemporary thermomechanical ice-sheet simulation we adopt is the Parallel Ice Sheet Model (PISM; Bueler and Brown (2009)) simulation of Aschwanden et al. (2016). This simulation represents the form, flow and thermal state of the Greenland ice sheet in c. 1990, following a 125 ky paleo-climatic spin-up, followed by another 2 ky of transient equilibrium with mass-flux adjustment forcing to minimize misfit against observed ice-sheet thickness and extent. This ice-sheet simulation has a horizontal resolution of 900 m and a vertical resolution of 20 m. PISM uses an enthalpy scheme for the conservation of energy calculation, in order to accommodate heat transfer in both freezing and temperate ice (Aschwanden et al., 2012). While the spin-up is meant to approximate the Greenland Ice Sheet at a specific time slice (c. 1990), we are comparing this simulated thermal state with temperature profiles observed over a 70-year time span. Shallow profiles located close to the margin can experience significant changes in ice thickness and temperature on this time scale. [2]

At locations of observed temperature profiles, modeled vertical temperature profiles were extracted from the PISM simulation based on the nearest neighbor grid point. Only temperature profiles from the Greenland Ice Sheet were included in this analysis. The PISM vertical temperature profiles were transformed from height above bed, to normalized depth below surface, and linearly interpolated to the vertical resolution of the normalized depth field of the database. Further, at a few drill site locations, the modeled ice was too thin, and did not have enough vertical grid points to interpolate the modeled temperatures to a proper normalized depth axis. It was therefore necessary to exclude profiles from the analysis where the simulated ice thickness was less than 5 vertical grid points (50 m). Additionally, in cases of temporally-repeated observed profiles, only the most recent measurement was included in this analysis see Table 4.

Out of the 85 temperature profiles from the temperature database, 56 were ultimately suitable for inclusion in this comparative analysis of observed and modeled ice temperatures. The observed temperature profiles were divided based on three characteristic regimes. First, whether profiles were located in the accumulation area (16 profiles) or ablation area (38 profiles). Secondly, whether the basal thermal state of profiles were considered warm-bedded (44 profiles) or cold-bedded (9 profiles). Finally, whether profiles were located in high-strain regions (10 profiles) or low-strain regions (46 profiles). The strain rate characterization was subjectively assigned by the author team, where high-strain rate was defined for sites approaching channelized glacier flow. Figure 6 shows the observed temperature profiles for each of the three regimes.

275

**Page: 14**

**Table 4.**

|  | Surface mass balance regime | Basal thermal state regime | Ice dynamic regime |
|---|---|---|---|
| Initial Number of profiles | 16 accumulation
38 ablation
2 unknown | 9 cold bedded
44 warm bedded
3 unknown | 10 high strain rate
46 low strain rate |
| Total | 56 | 56 | 56 |
| Number of profiles after exclusions | 16 accumulation
31 ablation | 5 cold bedded
40 warm bedded | 10 high strain rate
38 low strain rate |
| Total | 47 | 45 | 48 |

[Figure]

**Figure 6.** Observed temperature profiles on normalized depth scale for three characteristic regimes. **(A)**: Surface mass balance regime with profiles located in the ablation area (pink) or the accumulation area (gray). **(B)**: Basal thermal state regime with profiles either cold-bedded (purple) or warm-bedded (green). **(C)**: Ice dynamic regime with profiles located either in high-strain rate regions (orange) or low-strain rate regions (blue).

Number: 1 Author: brice    Subject: Cross-Out  Date: 20/10/2022 21:41:28
The caption is not related to the table. Same as table 3 !!

[revised manuscript text omitted]

---

## Author Comment (AC1)

**Greenland and Canadian Arctic ice temperature profiles database**

**Response to Referee Comments on tc-2022-138**

Anonymous Referee #1

We thank you for the time and energy that you have invested in reviewing our community paper. Below, please find responses to all comments. Your comments are in colored and italic text, and our replies are in black and plain font style.

*(1) In Table 1, four measurement methods are presented. However, the digital sensor string and thermistor string are not mentioned in the text. It is better to explain more details of the two measurement methods.*

In the text, 'sensor string' includes both the digital and analog variants. We have revised the text to make this more clear.

*(2) In Figure 1, the drill site location in the green box is not shown in Figure 1A. It is better to show the Jakobshavn glacier.*

We do not fully understand this comment. All drill sites are shown in this summary figure. Subplot 1B, with the green border, shows the upstream outer region of Jakobshavn Glacier. The location of this subplot, with boreholes therein, is indicated by the green square in Fig 1A.

*(3) Line 95: Please check the ice thickness in Tuto_D-11 borehole, it looks from the Figure 2 that the ice sheet thickness is 200 ft, which is about 61 m.*

A large length of the thermistor string is exposed on the ice sheet surface. From zooming into Figure 2, its text says "* 44 feet of thermocouple string was exposed on the ice beside the tube". Full resolution figure can be found at https://github.com/GEUS-Glaciology-and-Climate/greenland_ice_borehole_temperature_profiles/tree/main/boreholes/Tuto_D-11 where we add our notes, "Depth from text interpreted to mean that ice surface starts at 44 ft on cable and bottom is at 200 ft on cable. This yields an ice thickness of 156 ft or 156*0.3048 = 47.5488".

*(4) In the database, it is better to presented the temperature measurement methods (e.g., type and accuracy of temperature sensors) and depth measurement methods (e.g., type and accuracy of encoder) for the readers to evaluate the uncertainty of data source.*

We agree that additional metadata fields regarding uncertainty would be desirable. However, we cannot easily compile the original measurement method and its accuracy for each borehole. Many historical products provide limited information on the temperature sensor, and none on the depth estimate method. Because we provide detailed information on the upstream (original) data sources, readers can still come up with their own uncertainty for any individual borehole if needed. We also provide guidance for assessing total measurement uncertainty, which includes all mentioned sources of uncertainty. Future versions of the database will likely have additional metadata fields added, including uncertainty, which will be populated through expert elicitation and described in a future database description article.

*(5) Line 210-220: the paper of V. Zagorodnov et al. presented more detailed disturbance uncertainty of mechanical drill and some discussion can be included in the manuscript. (Zagorodnov, V., Nagornov, O., Scambos, T. A., Muto, A., Mosley-Thompson, E., Pettit, E. C., & Tyuflin, S. (2012). Borehole temperatures reveal details of 20th century warming at Bruce Plateau, Antarctic Peninsula. The Cryosphere, 6(3), 675-686.)*

We now include a statement that the temperature disturbance caused by mechanical drilling with fluid-filled boreholes dissipates to the level of precision within five days, and include this citation.

*(6) Section 6: Please provide more details how the author determined surface mass balance regime, the basal thermal state regime and ice dynamic regime. A table with accumulation/ ablation rate, basal temperature and strain rate is preferred.*

We now more fully describe these selection criteria. More specifically, surface mass balance regime is determined by whether the borehole is located below the snow line, in the ablation area, or above snow line, in the accumulation area, in contemporary satellite imagery (Figure 1). The basal thermal state regime is based on whether the ice-bed interface is measured to be below the pressure-melting-point temperature (i.e. frozen), or not (i.e. temperate). In instances where the borehole does not each the bed, we extrapolate the basal thermal state where reasonable (i.e. FladeIsblink06 is likely frozen), or we list basal thermal state as "unknown" where the extrapolation distance seems unreasonable (i.e. CampVI_50 is unknown). Finally, ice dynamic regime is classified as high strain when the ice flow is channelized, and low strain when sites are located in sheet- or divide-flow.

*In addition, some technical errors should be corrected.*
*(1) Line 15: "the thermal state of the sheet" should be "the thermal state of the ice sheet".*

Fixed - "ice" was included in the sentence.

*(2) Line 20: "thermo-mechanical" or "thermodynamic" or "thermomechanical"? Descriptions should be consistent throughout the manuscript.*

We now use "thermo-mechanical" throughout the manuscript for consistency.

*(3) Line 25: Please check the sentence "borehole logging where a temperature sensor is moved up or down the borehole measuring either "continuously" as the probe moves down". Borehole logger is used only when moves down? or, it can be used when moving down or up.*

I have deleted the word "down" so now the sentence reads: "...borehole logging where a temperature sensor is moved up or down the borehole measuring either continuously as the probe moves or is stopped to measure at every depth known as 'stop-and-go'."

*(4) Line 25: "fiber-optic distributed temperature sensing", "Fiber optic distributed sensing string"? The hyphenation between fiber and optic should be consistent throughout the manuscript.*

Fixed - a dash has been added so the manuscript now consistently has "fiber-optic".

*(5) Figure 1: The units of Celsius should have the same format throughout the paper.*

Figure 1 does not contain any temperature units. We suspect this comment perhaps refers to Figure 2, which is a reprint of a figure from the original study containing the DYE-3 temperature data with units "degC" (Gundestrup and Hansen, 1984). We cannot modify the figure that we are reprinting. Elsewhere, we have ensured we use (°C), rather than [°C], throughout.

*(6) Section 4: There are two "Figure 1" in the first sentence of the section.*

Fixed - the extra "Figure 1" has been removed.

*(7) Line220 and 230: "hot-water-drilled borehole" or "hot-water drilled borehole"? The style should be consistent throughout the manuscript. I think it should be "hot-water drilled borehole".*

Fixed - the dash was removed in line 230, now it reads "hot-water drilled" as suggested making the style consistent throughout the manuscript.

*(8) Table 4: The caption of the table 4 is the same as the table 3.*

Fixed - the caption has now been updated to match table 4 instead of table 3. New caption: "Overview of the number of profiles in the three regimes before and after excluding profiles not usable for the model comparison analysis."

*(9) The style of the references should be consistent, for example, the first letter of each word in the title of references should be lowercase. Please carefully check your references.*

The over capitalized references have now been changed, so the reference style is consistent.

Please also note the supplement to this comment:
https://tc.copernicus.org/preprints/tc-2022-138/tc-2022-138-RC1-supplement.pdf

We have addressed all comments in this response.

---

## Author Comment (AC2)

**Greenland and Canadian Arctic ice temperature profiles database**

**Response to Referee Comments on tc-2022-138**

Brice Van Liefferinge (Referee #2)

We thank you for the time and energy that you have invested in reviewing our community paper. Below, please find responses to all comments. Your comments are in colored and italic text, and our replies are in black and plain font style.

*Comment from feedback overview:*
*The xlsx file is for me unnecessary as the dataverse and github repository are well defined and clear. I would like to emphasize the quality of the open data sets shared with this Publication.*

The .xlsx file only lists the borehole classification that we adopt for Section 6 of this manuscript (comparison with PISM temperatures). We therefore only link this classification .xlsx to this manuscript, rather than the Dataverse/GitHub, as these classifications may vary with user interpretation.

*Page 1*
*L1: In the xlsx supplement 70 boreholes are identified but i see 79 on github and dataverse*

The supplement file has now been updated to include categorization for all 95 borehole profiles. Note, additional temperature profiles have been included in the database since the referee comment was posted.

*L2: replace "profiles" with "boreholes data". If you use "profiles", it refers to the "85 ice temperature profiles" and as a result, the percentage should be 29.5%*

We appreciate the potential for confusion, however, instead of using "boreholes data" we have rephrased the sentence in the following way "Profiles from only 31 boreholes (36%) were previously available in open-access data repositories". We hope this eliminates the confusion. Note, the number of borehole profiles and the percentage of the database it makes up has changed since the referee comments were posted, since additional profiles have been included in the database.

*L3: same here, it should refer to the boreholes and not the profiles*

Same here - we rephrased the sentence: "The remaining 54 borehole profiles (64%) are being made digitally available here for the first time."

*L5-6: these two values must be included in the xlsx supplement.*

The .xlsx file only lists the borehole classification that we adopt for this specific manuscript discussion point (Section 6; comparison with PISM temperatures). The database is not available in .xlsx format. It is only available in .csv format for both the Dataverse DOI snapshot and the GitHub living database. The two ice thickness scales are already available in both these database versions.

*L18: ice*

Fixed

*L36: e.g.*

Fixed - e.g. is added as suggested.

*L39-40: could you please provide more details: locations, depth, how many did you find,...? Too vague*

We know from historical accounts that previous profiles exist but locating these data has not been straightforward. We have no constraint on the number of pre-1950 profiles.

*L39: delete "may"*

For the above mentioned reason we decided to keep the "may" in the sentence.

*fig1: I don't really understand why use a satellite image as background as it is never referred to in the main document. It might be more useful to have something useful for the discussion e.g. a simpler colored background with grounding line indicated, zones of surface melt, …*

We now refer to using satellite imagery to classify whether boreholes are located above or below the contemporary snow line as an indicator of whether their surface mass balance regime is accumulation or ablation.

*Your KML file is in the EPSG 4326 projection. As you use the EPSG 3413 projection for the figures, it would be good to add a Geojson or .shp file with these projections in the repository. It's a plus not a must have*

Done. See https://github.com/GEUS-Glaciology-and-Climate/greenland_ice_borehole_temperature_profiles/commit/8afaa2014d2c2790fc6a4191276eb7d23a2976fa . Uploaded to GitHub for now. This has also been uploaded to dataverse, but we are waiting to publish the next Dataverse version when multiple changes associated with this review round have been accepted.

*Why did you apply a cubic spline interpolation? Why not simply use a simple piecewise linear interpolation ? The effects of thermodynamical parameters (mainly surface temp, geothermal heat flow, ice velocity, ...) on the ice temperature vary with depth and so a cubic spline is not necessarily the best fit for all depths. Could you explain why you used a cubic spline everywhere ?*

We opted for cubic spline because there are some records with large depth distance between measurements and a curved line appeared to fit the data better. We provide easy access to both the raw data (see `data.csv` file in each folder in https://github.com/GEUS-Glaciology-and-Climate/greenland_ice_borehole_temperature_profiles/tree/main/boreholes ) and our code so users can access raw data if they need to reprocess it differently.

*L78:  cross out "format" behind "KML"*

Fixed

*L78:  insert "format" in front of "KML"*

Fixed

*L85: insert "t"*

Fixed

*Page 7*
*I understand that it is complex to show all the drill site temperature data in one figure but at this scale it is difficult to see which data set is which. In itself, the figure is interesting to show the overall variations in the temperature data but not useful to show specific data sets. Maybe using differing linestyles would help separating out data sets visually. Another solution would be to make a bigger figure as supplementary material.*

We have tried many different visualization approaches pre-submission within our author team including some of the points suggested here, but all seemed to have shortcomings. So we are receptive to specific editorial feedback. To accommodate the comment, the figure has been made larger but is currently still included in the main text.

*Page 10*
*You should mention how deep you find the coldest temperature (surface, near the surface). It could give the false impression that the whole profile is quite cold which is only true in the upper part of the profile and does not reflect the processes occurring near the bed*

I have added the following "halfway down the profile at normalized depth 0.48." So the full sentence now reads: "The inland profiles also exhibit the coldest temperatures, with temperatures reaching a minimum of around -32°C halfway down the profile at normalized depth 0.48"

*Page 13*
*Table 3: Could you please define what "negligible" is. For some, 0.01 °C is also negligible (e.g. modelers). In other words, what is your defined limit ?*

We now state "negligible (<0.01 °C)" in the table caption to constrain the adjective for readers.

*Section 6: This paragraph is interesting and shows the importance of providing good constraints for models. You should add a table listing the key parameters used in the model (even-though it is not a paper on PISM) and explain how well they are constrained.*

Instead of making a table we added the following (at the end of section 6): "In this modeled run key parameters influencing the modeled temperatures are the enthalpy, which requires understanding of the liquid water fraction in the ice column and its evolution in time, the geothermal heat flow which also have the potential to affect the presence of water in a given location, and the surface boundary conditions i.e. accumulation rate, which control vertical temperature advection, and ice surface temperatures. These parameters are all very difficult to constrain, however, this is especially the case for the enthalpy and the geothermal heat flow parameter. "

*L250: You should mention that GHF is underestimated in some locations, see the paper of :
Rezvanbehbahani, S., Stearns, L. A., Kadivar, A., Walker, J. D., and van der Veen C. J.
(2017). Predicting the geothermal heat flux in Greenland: A machine learning approach.
Geophysical Research Letters, 44, 12,271–12,279. https://doi.org/10.1002/2017GL075661*

We now acknowledge that there is a diversity of opinion regarding the magnitude and spatial
distribution of geothermal heat flow beneath the ice sheet, and that modeled ice
temperatures are likely influenced by choice geothermal heat flow map. We cite
Rezvanbehbahani et al. (2017) and Colgan et al. (2022), which provide end members.

*L259: The constraint at the bed (GHF) has a key influence on the thermal state. Could you
please develop in one or two sentences the conditions used in the model at the bed?*

Yes - We have added the following sentence "The geothermal heat flow map by Shapiro and
Ritzwoller (2004), variable in space but not time, was used as a basal thermal boundary
condition."

*Table 4: the caption is not related to the table. Same as table 3 !!*

The caption has now been updated to match table 4 instead of table 3. New caption:
"Overview of the number of profiles in the three regimes before and after excluding profiles
not usable for the model comparison analysis."

*L302: The community has decided to use flow and not flux anymore see :
https://tc.copernicus.org/articles/14/3843/2020/ and the associated white paper. Please
change flux to flow in the whole manuscript.*

The term "flux" has now been changed to "flow" in the four instances where it occurred in the
text.

*L306: Please also mention the paper of Rezvanbehbahani, S., et al, 2017 see above*

We have now cited this paper.

*L441: the link is not valid*

Fixed, the link is now valid.

---

## Referee Report (RR1)

**Comments on tc-2022-138-Revised-Version**

After revision, the manuscript has been sufficiently improved with minor corrections need to be confirmed.

(1) Line 2: In the response to reviewers, the sentence is "*Profiles from only 25 boreholes (32%) previously available in open-access data repositories*", while it is "*Profiles from only 31 boreholes (36%) were previously available in open-access data repositories*" in the revised manuscript. The value should be 31, is it right?

(2) Line 26: In the response to reviewers, the word "down" has been deleted, however, it was kept in the revised manuscript.

(3) Line 284: Change "each" to "reach".

---

## Author Response (AR2)

**Greenland and Canadian Arctic ice temperature profiles database**

**Response to Comments on tc-2022-138-Revised-Version**

**Anonymous Referee #1**

We thank you for taking the time to review the manuscript again. Below, please find responses to all comments. Your comments are in colored and italic text, and our replies are in black and plain font style.

*(1) Line 2: In the response to reviewers, the sentence is "Profiles from only 25 boreholes (32%) previously available in open-access data repositories", while it is "Profiles from only 31 boreholes (36%) were previously available in open-access data repositories" in the revised manuscript. The value should be 31, is it right?*

Yes the value should be 31, it is written correctly in the manuscript.

*(2) Line 26: In the response to reviewers, the word "down" has been deleted, however, it was kept in the revised manuscript.*

It has now been removed from the manuscript.

*(3) Line 284: Change "each" to "reach".*

Done